



# 3-D tomographic reconstruction of atmospheric gravity waves in the mesosphere and lower thermosphere (MLT)

Rui Song [1,2], Martin Kaufmann [1], Manfred Ern [1], Jörn Ungermann [1], Guang Liu [3], and Martin Riese [1,2]

[1]Institute of Energy and Climate Research, Stratosphere (IEK-7), Research Centre Jülich, 52425 Jülich, Germany
[2]Institute for Atmospheric and Environmental Research, University of Wuppertal, 42119 Wuppertal, Germany
[3]Key Laboratory of Digital Earth Sciences, Institute of Remote Sensing and Digital Earth, Chinese Academy of Sciences, Beijing, China

*Correspondence to:* Rui Song (r.song@fz-juelich.de)

**Abstract.** Gravity waves (GWs) have been intensively studied over recent decades because of their dominant role in the dynamics of the mesosphere and lower thermosphere (MLT). The momentum deposition caused by breaking GWs determines the basic structure and drives the large-scale circulation in the MLT. Satellite observations provide a way to qualify the properties and effects of GWs on a global scale. As GWs can propagate vertically and horizontally in the atmosphere, resolving

both horizontal and vertical wavelengths is important for the quantification of a wave. However, this can hardly be achieved by one instrument with a good spatial coverage and resolution. In this paper, we propose a new observation strategy, called 'sweep mode', for a real three-dimensional (3-D) tomographic reconstruction of GWs in the MLT by modifying the observation geometry of conventional limb sounding measurements. It enhances the horizontal resolution that typical limb sounders can achieve, while at the same time retaining the good vertical resolution they have. This observation strategy is simulated

for retrieving temperatures from measurements of the rotational structure of the $O_2$ A-band airglow. The idea of this observation strategy is to sweep the line-of-sight (LOS) of the limb sounder horizontally across the orbital track during the flight. Therefore, two-dimensional (2-D) slices, i.e. vertical planes, that reveal the projection of GWs can be observed in the direction along- and across the orbital track, respectively. The 3-D wave vector is then reproduced by combining the projected 2-D wave slices in the two directions. The feasibility of this 'sweep mode' tomographic retrieval approach is assessed using simulated

measurements. It shows that the horizontal resolution in both along- and across-track directions are affected by an adjustable turning angle, which also determines the spatial coverage of this observation mode. The retrieval results can reduce the errors in deducing momentum flux substantially by providing an unbiased estimation of the real horizontal wavelength of a wave.

## 1 Introduction

The mesosphere and lower thermosphere (MLT, ∼50-110 km) constitutes the upper part of the middle atmosphere (∼10-110

km), and is dominated by atmospheric waves including planetary waves, tides and gravity waves (GWs) (Vincent, 2015). GWs are mainly excited in the lower atmosphere and propagate into the middle and upper atmosphere. As GWs propagate upward and dissipate, they force the large-scale atmospheric circulation, along with considerable influences on the constituents and thermal structures in the MLT region (Lindzen, 1981; Fritts and Alexander, 2003). Parameterizations of these processes are





important for reducing the uncertainty in weather and climate prediction models (Alexander et al., 2010; Geller et al., 2013). However, current parameterization schemes are based on simplified assumptions and lack constrains (Medvedev and Klaassen, 2000), resulting in an unrealistic assessment of the effects of GWs. Therefore, various observing techniques have been applied over recent decades to validate the GW parameterization models.

GWs can be observed from ground or space via the temperature perturbations that they cause in the atmosphere. Characterization of GWs from such temperature measurements first requires the subtraction of a background temperature. To this background temperature, the average temperature structure of the atmosphere contributes, as well as several different modes of planetary waves (Ern et al., 2009), and tides (Forbes et al., 2006). The remaining temperature perturbations are then assumed to be introduced by GWs. Ground-based observation techniques like, e.g. MF radars (e.g., Hoffmann et al. 2010, 2011), me-

teor radars (e.g., Fritts et al. 2010; Placke et al. 2011) and airglow imagers (e.g., Nakamura et al. 1999; Pautet and Moreels 2002; Suzuki et al. 2004, 2010) give information about the local time variations of GWs. Advances in satellite instrumentation have made it possible to measure GWs from a global perspective, allowing for identification of their spectral characterization and geographic distribution (e.g. Preusse et al. 2002; Wu 2004; Hoffmann and Alexander 2009; Ern et al. 2004, 2017). However, many of the spaceborne instruments are limited either in vertical or horizontal resolution due to their viewing geometry

(Alexander and Barnet, 2007).

Limb sounding is used in many satellite missions because it provides important information about the middle atmosphere dynamics with a good vertical resolution (∼1-3 km). Limb Infrared Monitor of the Stratosphere (LIMS) gave temperature profiles between 15 and 60 km at a vertical resolution of 1.8 km (Remsberg et al., 2004). Utilizing LIMS data, Fetzer and Gille (1994) derived global GW temperature variances with vertical wavelengths of ∼6-50 km and horizontal wavelengths longer

than ∼200 km. Global Positioning System (GPS) signals can be used to derive the temperature in the altitude range of ∼15-40 km with the radio occultation technique (Rocken et al., 1997). The derived temperature perturbations have been used by Tsuda et al. (2000) to extract mesoscale GWs with vertical wavelengths of ∼2-10 km. The Cryogenic Infrared Spectrometers and Telescopes for the Atmosphere (CRISTA) delivered temperature data from ∼20 to 80 km (Riese et al., 1999). From these data GWs with vertical wavelengths of ∼5-25 km and ∼6-30 km were derived from CRISTA-1 and CRISTA-2, respectively

(Eckermann and Preusse, 1999; Preusse et al., 2002). With its high vertical resolution (<1 km), the High Resolution Dynamics Limb Sounder (HIRDLS) data gave information about GWs with vertical wavelengths of ∼2-16 km in the altitude range of ∼20-60 km (Alexander et al., 2008; Ern et al., 2011). Compared to limb soundings, nadir soundings have a better horizontal resolution, but suffer from a poor vertical resolution. They include the Advanced Microwave Sounding Unit (AMSU) (Wu, 2004) and the Atmospheric Infrared Sounder (AIRS) (Alexander and Barnet, 2007; Hoffmann and Alexander, 2009; Ern et al.,

30  2017).

For a better understanding of GW effects, proper observations with both high vertical and horizontal resolutions are required. Typical limb sounders can resolve GW structures with a high vertical resolution by assuming a horizontally homogeneous atmosphere along the line-of-sight (LOS) of the observations. In the case of a spaceborne platform, the effective length of an atmospheric parcel (i.e., the weighting function) that is scanned by a limb-viewing instrument can extend to a length of

a few hundred kilometers along the LOS. This is a major error source for the retrieval of the true state of the atmosphere,





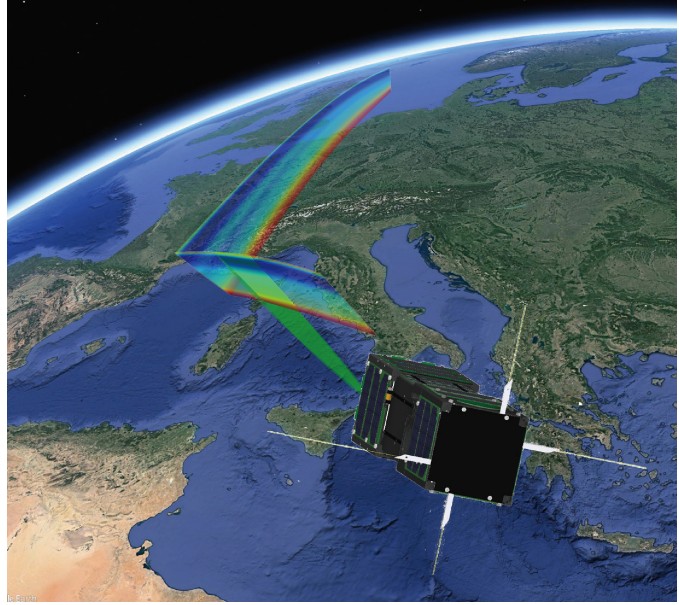

**Figure 1.** Observation geometry of a nanosatellite operated in 'sweep mode'. The two slices represent the atmospheric temperature structures in the along- and across-track directions, respectively.

especially when the probed region suffers from strong variabilities, e.g. GWs. However, large part of horizontal variabilities of the atmosphere can still be accounted for from limb sounding measurements by means of a modified retrieval scheme. The approach of obtaining horizontal structures of GWs applies to the general limb instruments with a fixed viewing angle with respect to the flight track. The basic idea of this approach is to estimate the horizontal wavelength of GWs by combining the

phases provided by the wave analysis of adjacent vertical profiles. This approach was proposed by Ern et al. (2004) and was successfully used to retrieve wave temperature amplitudes, vertical wavelengths and projections of horizontal wavelengths in the sampling direction with CRISTA-2 measurements, as well as several other datasets (Alexander et al., 2008; Wright et al., 2010; Ern et al., 2011). The shortest horizontal wavelength to be observed is limited by the Nyquist wavelength, which is twice of the satellite sampling distance. If the LOS is in the direction of the orbital plane, horizontal inhomogeneities along

the LOS can be obtained more reliably by applying a 2-D tomographic retrieval scheme. This approach has been used in Michelson Interferometer for Passive Atmospheric Sounding (MIPAS) (Carlotti et al., 2001) and Microwave Limb Sounder (MLS) (Livesey and Read, 2000). However, even if the horizontal resolution can be improved by such approaches, there is one main limitation that applies for all observations based on a fixed LOS: only the apparent wavelengths along the orbital track are derived. This generally results in a retrieved horizontal wavelength longer than the real wavelength of a wave.

Therefore, deriving highly-resolved three-dimensional (3-D) temperature fields would represent a major advance. It would ensure that the resolved GW structures are realistic in terms of wavelength, amplitude, and propagation direction. A real 3-D tomographic retrieval combining different limb-scanning sequences requires a large number of tangent points in the target



atmospheric volume. In the case of an airborne platform, this can be realized by viewing the target volume from different directions, for example by performing closed flight patterns that enclose the target volume, or by panning the viewing direction of the instrument during flight. Such an observation scenario was suggested by Ungermann et al. (2011) and applied by Kaufmann et al. (2015) and Krisch et al. (2017) for an airborne infrared limb sounder. For a spaceborne platform, the feasibility

of resolving fine GW structures with tomographic retrievals has been demonstrated using simulated measurments of PREMIER mission (Process Exploration through Measurements of Infrared and millimetre-wave Emitted Radiation)(Ungermann et al., 2010). The PREMIER concept is based on infrared limb-imaging (Riese et al., 2005), which provides high along-track sampling (∼50 km) and across-track sampling (∼25 km) at the same time by combining novel two-dimensional detector arrays with Fourier spectroscopy (e.g., Friedl-Vallon et al. 2014).

In this paper, we present a new spaceborne observation strategy to detect 3-D atmospheric structures in the MLT region. It is applicable to any temperature limb sounder with vertical imaging capability and ability to change its viewing direction rapidly as shown in Fig. 1. The idea of this observation mode is to locate the tangent points in 2-D slices by sweeping the LOS of the instrument in the horizontal plane. The reconstruction of the 3-D atmospheric state is then simplified by performing retrievals of 2-D atmospheric slices that represent the projections of the true atmosphere in two different directions. The observation

strategy of this 'sweep mode' is introduced in Sect. 2 in more detail. The forward model for this 3-D tomographic retrieval problem is described in Sect. 3. In Sect. 4, a case study of this 'sweep mode' using simulated measurements is presented. The achievable horizontal resolution and coverage, as well as the unbiased estimation of the real horizontal wavelength of a wave from this observational mode is analyzed in Sect. 5. Finally, the conclusion is given in Sect. 6.

## 2 Instrument and observation strategy

This study was designed for an instrument measuring the rotational distribution of the $O_2$ A-band emission. The rotational distribution of the emission follows a Boltzmann distribution described by the kinetic temperature. Kaufmann et al. (2017) describe the concept of an instrument based on current nano-satellite platforms that is capable to measure this emission. Depending on the signal-to-noise requirement, this instrument delivers temperature at a vertical resolution of a few hundred meters.

The high-performance attitude control system provides the instrument with the ability to sweep the LOS accurately. While conventional spaceborne limb sounders usually have a constant viewing direction during the flight, this 'sweep mode' is capable of adjusting the LOS from forward looking to backward looking. During the sweeping, a slew rate of ∼10 deg/sec provided by the envisaged attitude control system is sufficient for the purpose of sampling the vertical atmospheric profiles along- and across the orbital track, respectively. Fig. 2 illustrates the orbital track, flight direction, satellite positions and corresponding

tangents points for conventional limb sounding and 'sweep mode' limb sounding, respectively. In this simulation, the satellite is operated in an orbit at an altitude of 600 km and inclination of 98°. The satellite positions are represented by green dots, and corresponding tangent points are represented by red triangles. The flight direction is indicated by the blue arrows along the orbital track, and the direction of one sample LOS is indicated by the black arrow. The viewing angle, defined as the angle





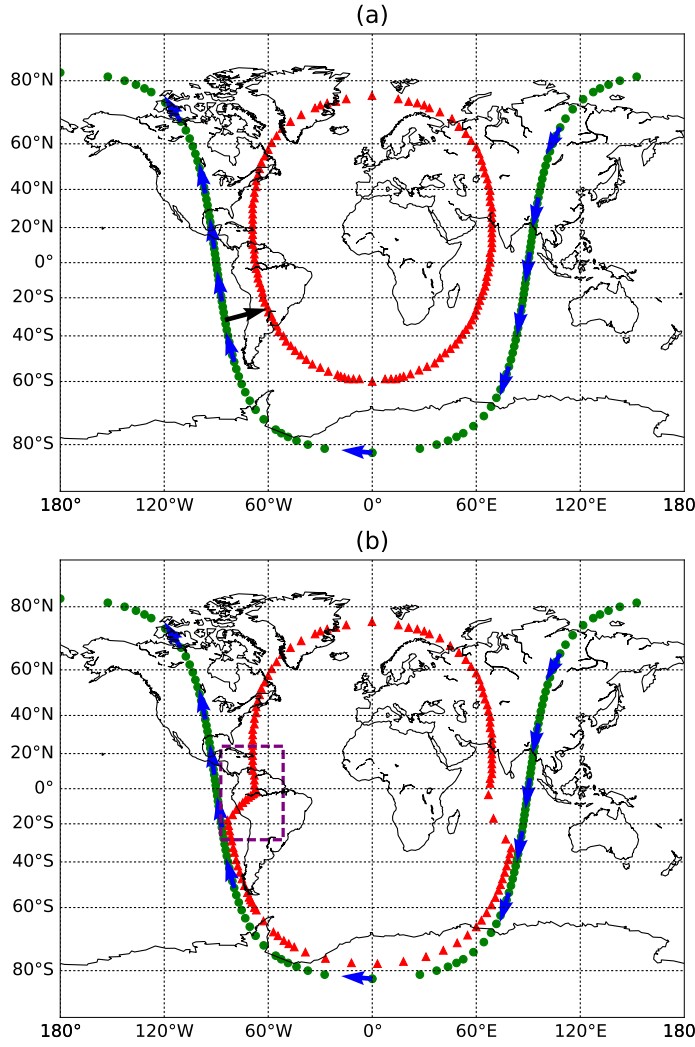

**Figure 2.** Global observation geometry of an exemplary orbit of (a) conventional limb sounding and (b) 'sweep mode' limb sounding. The satellite is operated in an orbit at an altitude of 600 km and inclination of 98°. Satellite positions are sampled by green dots and corresponding tangent points by red triangles. The flight direction of the satellite is shown by blue arrows. The viewing angle in panel (a) is perpendicular to the flight direction, shown as the black arrow. In panel (b), the viewing angle changes from 10° to 90° in the ascending orbit, and decreases back to 10° in the descending orbit.





between flight direction and LOS, is a constant value for conventional limb sounding, e.g. 90° as shown in Fig. 2a. While the viewing angle for 'sweep mode' needs to be adapted in a way such that the two vertical slices are exactly perpendicular to each other. Fig. 2b shows an example of 'sweep mode' with the viewing angle of the instrument increasing from 10° to 90° in the region marked by the purple-dashed rectangular box. In this region, a 2-D retrieval scheme is used to resolve small-scale

5 atmospheric structures in the direction along- and across the orbit track separately. Consequently, the 3-D GW parameters within this atmospheric volume can be derived by combining the 2-D wave vectors along the two directions mentioned above.

## 3 Forward model

### 3.1 $O_2$ A-band ariglow emission model

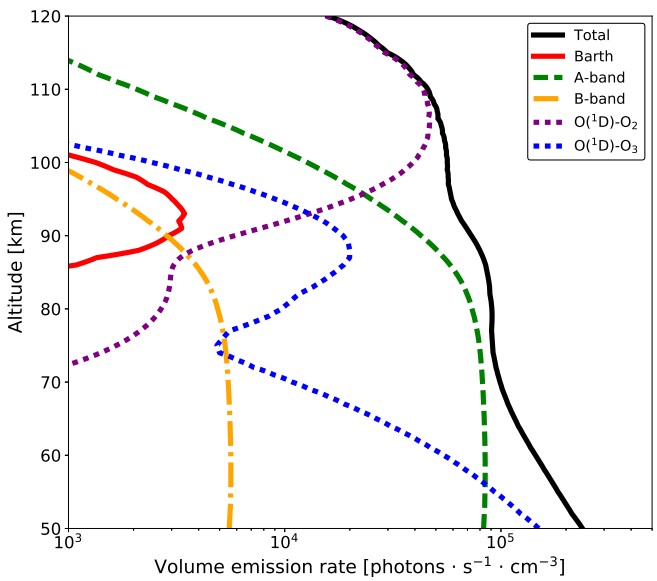

**Figure 3.** Modeled $O_2$ A-band dayglow emission profile at 30° N and 220° E for 14:40 local solar time simulated from the HAMMONIA model. A-band and B-band (dashed green line and dot-dashed orange line, respectively) are the contributions from the resonant absorption. $O(^1D)$-$O_2$ and $O(^1D)$-$O_3$ (dotted purple line and blue line, respectively) are the contributions from the collision with $O(^1D)$. The solid red line is the contribution from the Barth process. The black curve denotes the total A-band emission.

The oxygen atmospheric band (A-band) airglow emission is used for the temperature measurements within the required altitude

10 range (∼60-120 km). This oxygen band emission is self-absorbed such that radiance from the lower atmosphere or the ground cannot reach the spaceborne instrument. In the MLT region the $O_2$ A-band becomes optically thin, making it possible to measure this emission from space. There are three primary sources of the excited oxygen state $O_2(b^1\Sigma, v = 0)$ in the MLT region, which lead to the A-band emission. The first source is through the A-band (762 nm) and B-band (689 nm) resonant





absorption from the ground state $O_2(X^3\Sigma, v=0)$:

$$O_2(X^3\Sigma, v=0) + h\upsilon \rightarrow O_2(b^1\Sigma, v=0,1) \tag{1}$$

The second source is through the collision of $O_2(X^3\Sigma, v=0)$ with $O(^1D)$:

$$O_2(X^3\Sigma, v=0) + O(^1D) \rightarrow O(^3P) + O_2(b^1\Sigma) \tag{2}$$

The third source of $O_2(b^1\Sigma)$ is a two-step Barth process, in which an excited molecule $O_2^*$ is created from atomic oxygen first, and afterward $O_2(b^1\Sigma)$ is produced by quenching.

$$O + O + M \rightarrow O_2^* + M$$
$$O_2^* + O_2 \rightarrow O_2(b^1\Sigma) + O_2 \tag{3}$$

The Barth process is independent of solar radiation, and therefore is the only mechanism that produces $O_2(b^1\Sigma, v=0)$ during nighttime. Fig. 3 shows a typical vertical profile reflecting the volume emission rate of modeled $O_2$ A-band dayglow, along

with the relevant excitation processes described above. The background atmosphere (temperature T, number density of O, $O_2$, $O_3$ and $N_2$) is taken from the Hamburg Model of the Neutral and Ionized Atmosphere (HAMMONIA) model (Schmidt et al., 2006) run at 30° N and 220° E for 14:40 local solar time. As shown in Fig. 3, the $O_2$ A-band volume emission rate $\eta$ has a broad maximum of $\sim 10^5$ photons $\cdot$ s$^{-1}$ $\cdot$ cm$^{-3}$ from 50 to 100 km, with exponential decrease above that. Since the $O_2(b^1\Sigma, v=0)$ has a lifetime of more than 12 sec, it is generally assumed that the molecule is in rotational local-thermodynamic equilibrium

(Vallance Jones, 1974). Therefore, the kinetic temperature of the atmosphere can be derived from the rotational band structures of the emissions. As a consequence of the rotational band structure, the number of emitted photons is distributed into narrow emission lines within the band, represented by $\eta_{\text{rot}}$. At a rotational temperature $T$, the rotational band structure follows the Boltzmann distribution, and detailed calculation can be found in Mies (1974). Within this airglow emission spectrum, a subset of six emission lines has proven to give an optimal setup for a potential satellite mission aiming to the derivation of kinetic

temperature. The physical parameters used for this fine structure calculation are provided by the high-resolution transmission molecular absorption database (HITRAN) (Rothman et al., 2013) and listed in Table 1. They include wavenumber $\upsilon$, lower state energy $E''$, Einstein coefficient $A'$, and upper state degeneracy $g'$. The lines were chosen so that both positive and negative temperature dependency of rotational structures are distinct. Plots of normalized rotational structure as a function of temperature are shown in Fig. 4.

## 3.2   Wave perturbation

The temporal and spatial variability of the airglow emission are affected by the changes in constituents and temperature due to upward propagating GWs. Considering an adiabatic and windless atmosphere, the temperature perturbation $T'$ at position $(x, y, z)$ induced by a monochromatic wave can be written as (Fritts and Alexander, 2003):

$$T'(x,y,z,t) = \hat{T} \cos\left(\frac{2\pi x}{\lambda_x} + \frac{2\pi y}{\lambda_y} + \frac{2\pi z}{\lambda_z} - \hat{\omega}t\right) \tag{4}$$



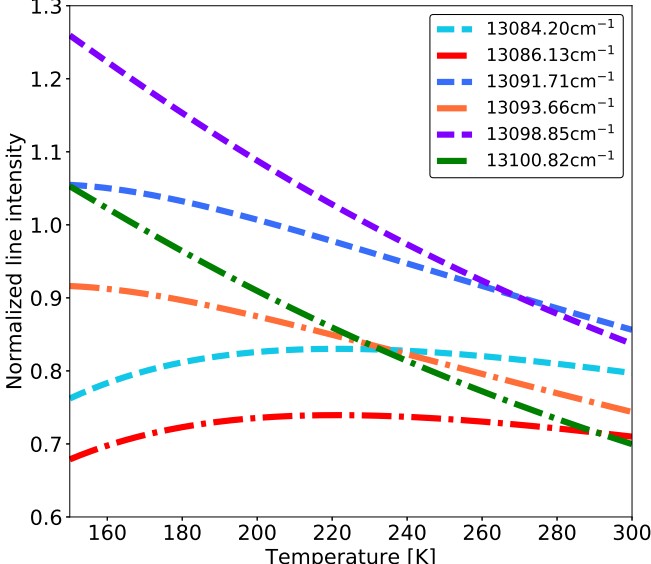

**Figure 4.** Temperature dependence of six rotational lines of the O$_2$ A-band emission. The center wavenumbers of individual lines are given in the figure legend. The intensity is normalized around the maximum intensity for a temperature of 230 K.

**Table 1.** Constants for O$_2$ A-band rotational structure calculation

| $\upsilon$, cm$^{-1}$ | $E''$, cm$^{-1}$ | $A'$, s$^{-1}$ | $g'$ |
|---|---|---|---|
| 13084.203 | 190.775 | 2.506$e$-02 | 21.0 |
| 13086.125 | 188.853 | 2.232$e$-02 | 21.0 |
| 13091.710 | 130.438 | 2.581$e$-02 | 17.0 |
| 13093.656 | 128.492 | 2.242$e$-02 | 17.0 |
| 13098.848 | 81.581 | 2.701$e$-02 | 13.0 |
| 13100.822 | 79.607 | 2.258$e$-02 | 13.0 |

where $\hat{T}$ is the wave amplitude, $\hat{\omega}$ the intrinsic frequency, $\lambda_z$ the vertical wavelength, $\lambda_x$ and $\lambda_y$ the horizontal wavelengths in $x$ and $y$ directions, respectively. The number densities of various constituents that contribute to the O$_2$ A-band airglow emission are also affected by the GW. Ward (1999) proposed a model to simulate the perturbations of airglow taking into account the effects from both the temperature and relevant constituents. The GW induced perturbation on the O$_2$ A-band airglow emission at position $(x, y, z)$ can be calculated with this model.



### 3.3 Radiative transfer

The spectral irradiance $I(v)$ observed by the instrument is a path integral along the LOS, in photons$\cdot$s$^{-1}\cdot$cm$^{-2}$:

$$I(v) = \int_{-\infty}^{\infty} \eta(s)_{\mathrm{rot}}\, D(v,s) \exp[-\int_{-s}^{\infty} n(s')\, \sigma(s')\, D(v,s)\, ds']\, ds \qquad (5)$$

where $n$ is the O$_2$ number density and $\sigma$ is the absorption cross section. $s$ represents the propagation path through the atmo-
sphere to the instrument along the LOS. The Doppler broadening at wavenumber $v$ is characterized by the function $D(v)$.
This propagation path $s$ consists of a set of points at the crossings of the LOS with specified atmospheric grids, along with the
distance between neighboring points. As the satellite instrument is above the atmosphere, this geometrical calculation starts at
the point where the ray enters the top of the atmosphere, and ends where it leaves the top of the atmosphere. This path is then
tracked backwards for the calculation of radiation absorption as shown in Eq. 5. Atmospheric Radiative Transfer Simulator
(ARTS) is a free open-source programming software (Buehler et al., 2005), and it allows the simulation for ray tracing through
an 1-D, 2-D or 3-D atmosphere. In our study, we use ARTS for a fast 3-D ray tracing of the instrument LOS.
The O$_2$ A-band emission is self-absorbed through the atmosphere. Above the stratopause ($\sim$50 km), only the thermal Doppler
broadening $D(v)$ at each emission line is significant and considered for the absorption calculation:

$$D(v) = \frac{1}{\alpha_D \sqrt{\pi}} \exp\left[ -\frac{(v-v_0)^2}{\alpha_D^2} \right] \qquad (6)$$

where $v_0$ is the central wavenumber of individual emission lines. The Doppler half-width at half-maximum is $\alpha_D \sqrt{\ln 2}$, with
$\alpha_D$ defined as:

$$\alpha_D = v_0 \sqrt{\frac{2 k_B T}{m c^2}} \qquad (7)$$

where $c$ is the speed of light, $k_B$ is Boltzmann's constant, $m$ is the weight of molecular oxygen, and $T$ is the atmospheric
temperature at position $s$. Instead of a regular wavenumber grid-spacing, an adaptive grid-spacing is used for the absorption
calculation in our study. Specifically, a finer grid-spacing is used in the range near the line center and a coarser grid-spacing is
used along the wings of the Doppler function. This adaptive grid-spacing reduces the computational cost, and is sufficient for
an accurate estimation of the atmospheric attenuation caused by absorption.
The irradiance received by the instrument is very dependent on the temperature and the O$_2$ density at the tangent altitude.
The temperature determines the total amount of airglow emission and the rotational structures, and the O$_2$ density determines
the self-absorption. Fig. 5 shows the irradiance response ($\Delta I(v)/I(v)$) to a temperature perturbation ($\Delta T$) of 10 K at tangent
altitudes ranging from 60 to 119 km with 1 km interval at individual emission lines. The constituents perturbation induced
by the $\Delta T$ is considered utilizing the model presented in Sect. 3.2. The irradiance response of the strong line (e.g. 13098.85
and 13100.82 cm$^{-1}$) to temperature perturbations is relatively weak ($<6\%$) for tangent altitudes below 87 km due to strong
self-absorption. Above 110 km, the sensitivity of the irradiance response to temperature perturbations reduces because of the
decrease in the total amount of O$_2$ A-band emission. Therefore, in this study the atmospheric temperature is retrieved between
87-110 km to ensure a moderate sensitivity to small temperature perturbations.





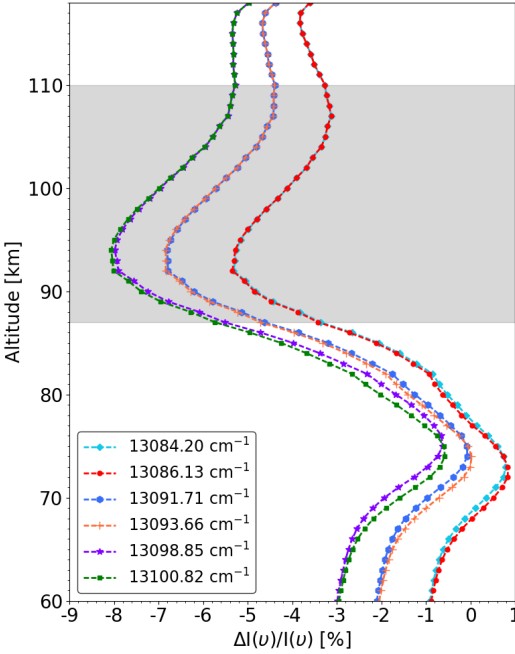

**Figure 5.** Response of irradiance ($\Delta I(\upsilon)/I(\upsilon)$) to temperature ($\Delta T = 10$ K) at different tangent altitudes for individual emission lines. The tangent altitudes covered by the grey shaded area (87-110 km) has a higher sensitivity of the irradiance to temperature changes.

## 4 Numerical experiments

The retrieval scheme adopted for solving this 3-D tomographic inverse problem has been summarized in Song et al. (2017). In this section, we use an example to show an end-to-end simulation of the 'sweep mode' tomographic observations. The capability of resolving 3-D wave structures from projected 2-D wave slices are explored. The parameters relevant for the observation geometry and atmospheric condition are introduced in Sect.4.1 and Sect.4.2, respectively.

### 4.1 Observation geometry setup

The simulation is based on a $\sim$600 km sun-synchronous orbit with an inclination angle of 98°. In this simulation, the tangent points are sampled with an altitude step of 1.5 km between $\sim$60-120 km, and the integration time is assumed to be 6 sec for $O_2$ A-band dayglow measurements. The measurement track and tangent point distribution of this 'sweep mode' are illustrated in Fig. 6. It provides a visual clue for the distribution characteristics of the tangents points in the 'sweep mode'. The solid lines represent the projection of the LOS near the tangent point, i.e., an altitude range 70-71 km, on a horizontal plane from consecutive measurements. The locations of corresponding tangent points are marked as black circles. In this example, the instrument is operated in the conventional mode with an viewing angle of 0° before the 'sweep mode' is activated. After activating the 'sweep mode', the instrument starts to increase its viewing angle from 0° towards a turning angle of 45°. Fig.



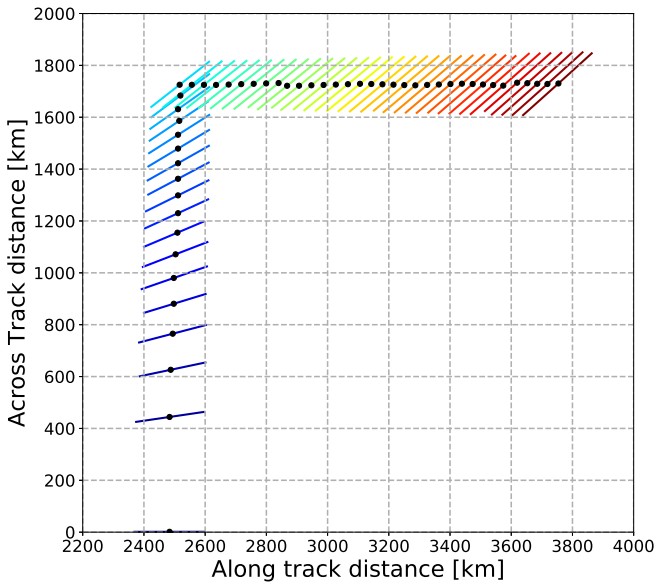

**Figure 6.** Illustration of the 'sweep mode' central measurement track. Flight direction is to the right, $0°$ viewing direction is to the right as well. For a single measurement, the LOS that targets at 70 km tangent altitude is traced. The solid lines indicate the segments of propagation paths between 70 km and 71 km in the atmosphere. The locations of corresponding tangent points are represented by black circles.

7 shows the viewing angle variations of the measurements taken in the 'sweep mode'. This 'sweep mode' is initiated with a viewing angle of $\alpha_0 = 0°$ and a slew rate of $\omega = \omega_0$ at $t_0 = 0$ sec. This slew rate slows down gradually such that the tangent points for each imaging are arranged perfectly perpendicular to the orbital track. At time $t$, the viewing angle of the instrument is

$$\alpha_t = \arccos(\frac{L - vt}{L}) \tag{8}$$

where $L$ is the distance between the tangent point and the satellite position, $v$ is the speed of the satellite. This viewing angle keeps increasing until the predefined turning angle of $45°$ has been reached at $t = 100$ sec. This viewing angle keeps increasing until the predefined turning angle has been reached. After that, the satellite operates again in the conventional mode with a newly set viewing angle of $45°$ in the subsequent orbital track.

## 4.2 Atmosphere setup

In this simulation, the forward model calculates radiance in a 3-D volume at tangent altitudes between 60 and 120 km on a 250 m vertical grid. In the horizontal direction, the spacing of the along- and across-track grid of the model is 5 km. This finely sampled atmospheric grid is essential for an accurate simulation of the radiation received by the instrument along the LOS. In the retrieval model, the temperature fields are estimated on two 2-D volumes: one in the across-track direction and another in the along-track direction. In the 2-D retrieval volume, the vertical spacing is 500 m and the horizontal spacing is 12.5 km.



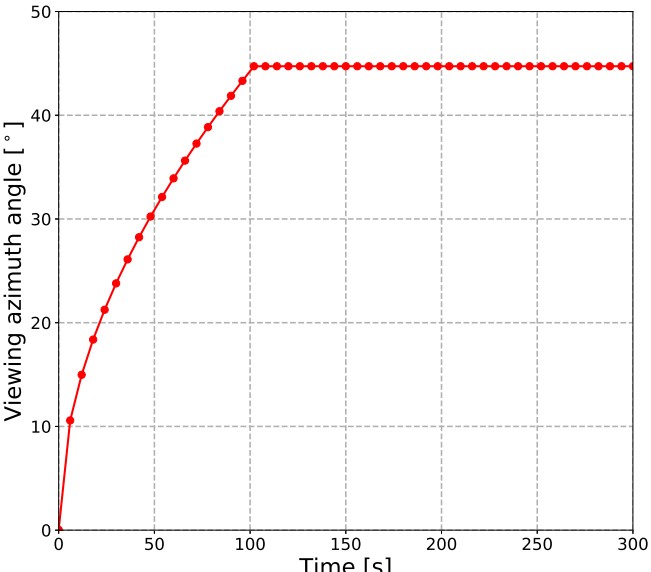

**Figure 7.** Viewing angle variations of simulated 'sweep mode' measurements.

## 4.3 3-D tomographic retrieval

This section describes the implementation of the presented principles of 'sweep mode' tomographic reconstruction in the numerical model. In this simulation, we examined the retrieval results by analyzing the extracted wave amplitude and wavelengths, with special attention paid to how well the horizontal wavelengths can be reproduced along the given directions.

To illustrate the performance of the new observation mode, we perturbed a background atmosphere with a GW, as shown in Fig. 8. The simulated GW has a vertical wavelength of 15 km and a horizontal wavelength of 310 km tilted at an angle of 38.6° away from the flight direction. The left panels in Fig. 8 represent the corresponding 2-D slices extracted from the 3-D atmosphere in the across-track direction. The a priori data, which is assumed to be the same as the background data, is depicted Fig. 8a. A GW induced temperature perturbation with an amplitude of 10 K (Ehard et al., 2015) is depicted in Fig. 8b. The locations of the tangent points from corresponding measurements are marked as white circles. As we can see, the tangent points have a smaller horizontal sampling distance in the region further apart from the orbital track. The retrieved temperature perturbation is shown in Fig. 8c. It is obvious that the retrieved wave has a smoother-looking wave pattern in the part with longer across-track distance, due to a denser horizontal sampling rate in this direction. To deduce the wavelength and amplitude of the retrieved GW, all atmospheric points are fitted against the modulated waves using a least-square best fit. In order to obtain optimal values, the vertical and horizontal wavelengths of the fitted sine wave are sampled at an interval of 0.1 km, and the amplitude is set as a free parameter for the best-fit calculating. After this best-fit processing, an optimized wave with a vertical wavelength of 15.2 km, an across-track wavelength of 396.6 km and an amplitude of 8.2 K is obtained, as shown in Fig. 8d. In contrast, the panels in the right side show the 2-D along-track slices of the background atmosphere, simulated, retrieved and fitted wave



**Figure 8.** Example of 'sweep mode' tomographic retrieval results using simulated data. The background atmosphere is taken from the HAMMONIA run from 134° W to 156° W and from 4° S to 19° N. The panels in the left hand side are 2-D slices extracted from the 3-D atmospheric volume in the across-track direction, while those in the right hand side are extracted from the along-track direction. The panels in the first row depict the a priori atmosphere. The panels in the second row show the simulated wave structures, along with the tangent points from individual measurements marked as white circles. The panels in the third row show the retrieved temperature perturbations. The panels in the last row depict the optimized wave structures after best-fit processing.



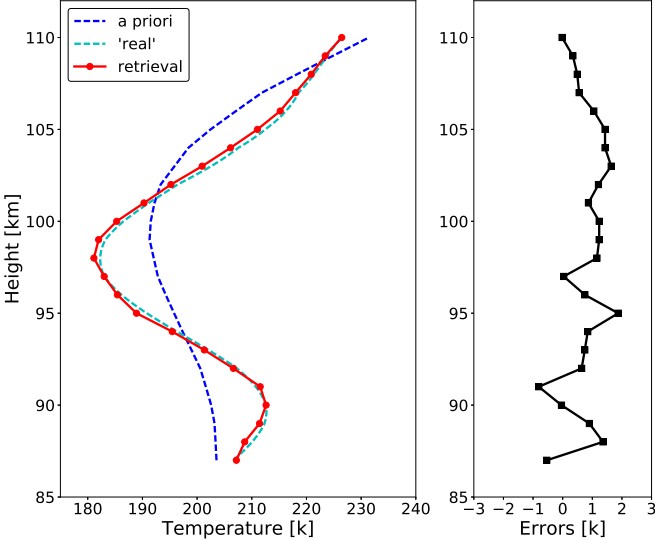

**Figure 9.** Vertical profile of retrieved temperature (left) and total deviation from the expected profile including all error sources (right).

structures, respectively. From Fig. 8f, one can clearly see consecutive measurements with a constant horizontal sampling rate in that direction. This results in a smooth retrieved wave structure over the entire region of interest, as shown in Fig. 8g. The optimized wave has a vertical wavelength of 15.1 km, an along-track wavelength of 498.2 km and an amplitude of 8.3 K.

Figure. 9 shows a vertical profile of retrieved temperature and total deviation from the expected profile including all error

sources at altitudes between 87 and 110 km. The error components considered in this simulation arise from the gridding error, the smoothing error, and mainly from the measurement noise. The temperature is retrieved with a precision better than 2 K within this altitude range. This uncertainty value allows for the derivation of small-scale atmospheric structures from the temperature perturbations.

The spatial response of the retrieval result is characterized by the averaging kernel matrix **A** given by:

$$\mathbf{A} = (\mathbf{S}_a^{-1} + \boldsymbol{f}'(\boldsymbol{x})^T \mathbf{S}_\epsilon^{-1} \boldsymbol{f}'(\boldsymbol{x}))^{-1} \boldsymbol{f}'(\boldsymbol{x})^T \mathbf{S}_\epsilon^{-1} \boldsymbol{f}'(\boldsymbol{x}) \tag{9}$$

where $\boldsymbol{f}'$ is the Jacobian matrix of the forward model $\boldsymbol{f}$ calculated at atmospheric state $\boldsymbol{x}$. $\mathbf{S}_a^{-1}$ is the regularization matrix used to constrain the solution, and $\mathbf{S}_\epsilon^{-1}$ is the covariance matrix of measurement errors. The spatial dispersion of the retrieved data point is measured by a single row of the averaging kernel matrix. Each element in the row describes how and where the retrieval is influenced by the true state of the atmosphere. Fig. 10 shows a row of the averaging kernel matrix after mapped into a 2-D

space according to the vertical and horizontal coordinates. The spatial resolution of the retrieved data point is estimated by calculating the full width at half maximum (FWHM) of the row of the averaging kernel matrix along the vertical and horizontal directions, respectively.

In Fig. 10, panel (a) shows the averaging kernel of the retrieval in across-track direction for a tangent point located at 96 km vertically and 1600 km across track. The estimated vertical and horizontal resolution are ∼1.2 km and ∼95 km, respectively.





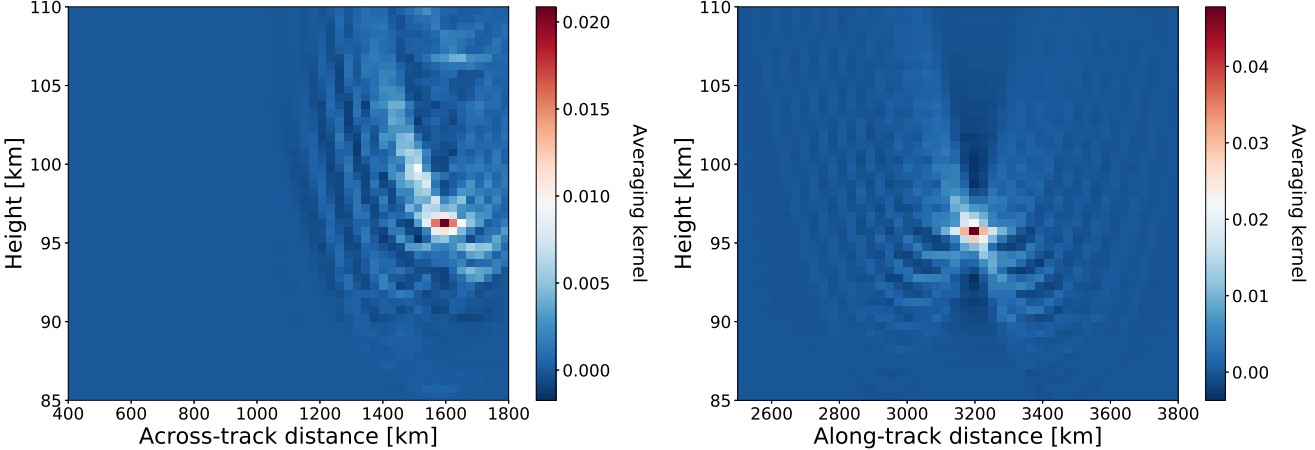

**Figure 10.** Averaging kernel matrix for retrieval points in the across-track slice (a) and along-track slice (b). Figure (a) is for the point located at 96 km vertically and 1600 km across-track, and Figure (b) is for the points locating 96 km vertically and 3200 km along-track.

The averaging kernel in the along-tack direction is shown in panel (b) at the position of 96 km vertically and 3200 km along track. The estimated vertical and horizontal resolution are ∼1.2 km and ∼80 km, respectively.

## 5 Horizontal wavelength analysis

GW momentum flux can be estimated from temperature measurements. Based on GW polarization relations, Ern et al. (2004) related GW momentum flux with retrieved temperature amplitude and vertical and horizontal wavelengths. Considering a medium-frequency wave ($N \gg \hat{\omega} \gg f$), the GW momentum flux is given by:

$$F = \frac{1}{2} \, \bar{\varrho} \, \frac{k}{m} \left( \frac{\mathrm{g}}{N} \right)^2 \left( \frac{\hat{T}}{T} \right)^2 \tag{10}$$

where g is the gravity acceleration, $N$ is the buoyancy frequency, $\hat{T}$ and $T$ are the temperature amplitude and background temperature, respectively. $k = 2\pi/\lambda_h$ is the horizontal wavenumber and $m = 2\pi/\lambda_v$ is the vertical wavenumber of the wave. Therefore, the momentum flux of a GW can be determined if the full wave vector and temperature amplitude are able to be retrieved from the measurements. The retrieval approach for the derivation of vertical wavelength and amplitude of a GW is well-established, and therefore further discussion is not given in the paper. In this section, we focus on the assessment of how well the horizontal wave vector can be reproduced from this 'sweep mode' tomographic retrieval.

### 5.1 Along-track horizontal wavelength

In Sect.4, the numerical experiments are simulated for the 'sweep mode' measurements with a specific turning angle of 45°. However, this turning angle $\alpha$ is flexible and can be adjusted between 0° and 90° according to the target area. Here, the performance of the along-track wavelength retrieval in two extreme cases of $\alpha = 0°$ and $\alpha = 90°$ are discussed.



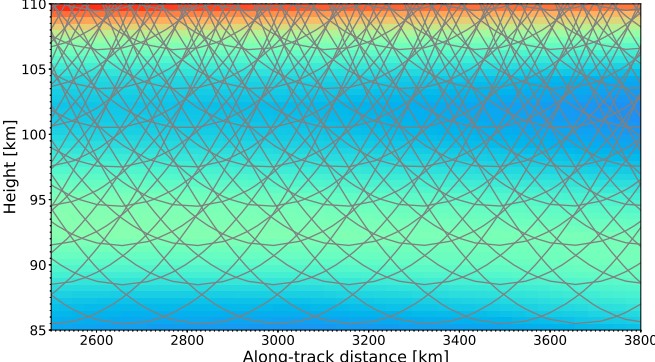

**Figure 11.** Central track of limb sounding measurements with a viewing azimuth angle of $0°$. Note that just for display in this figure the sampling distance is 3 km in the vertical and 165 km in the along-track direction, respectively. However, the simulated 'sweep mode' has a better spatial resolution: 1.5 km in the vertical and 40 km in the along-track direction.

When $\alpha = 90°$, the tangent points for along-track wavelength analysis extend to ∼2500 km apart from the orbital plane. For individual measurements, the LOS is perpendicular to the along-track direction in this case. This means all the radiation received by the instrument comes from the atmosphere in the across-track direction. For this viewing geometry, 1-D vertical temperature profiles are retrieved from individual measurements independently. The horizontal wavelength is then estimated

5   by analyzing the phases of adjacent vertical temperature profiles (Ern et al., 2004). In this case, the along-track horizontal wavelength that can be detected is limited by the Nyquist wavelength, which is twice the sampling distance: $\lambda_{h,N} = 2\Delta x_h$. Considering the integration time of ∼6 sec in this simulation, GWs with horizontal wavelength longer than ∼ 85 km in the along-track direction can be detected.

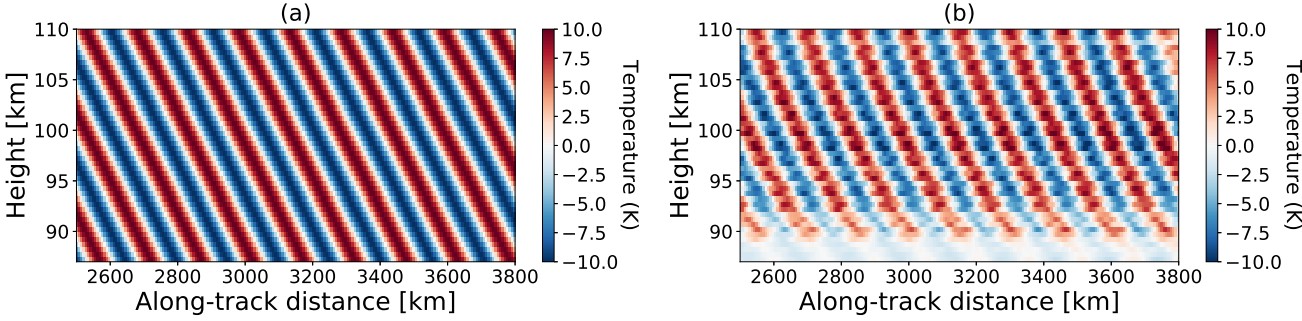

**Figure 12.** Simulated GW (a) with a vertical wavelength of 10 km and horizontal wavelength of 150 km, and retrieved wave structure (b) assuming a turning angle $\alpha$ of $0°$.

If the turning angle $\alpha$ is smaller than $90°$, more information from neighboring atmospheric volume in the along-track direction will contribute to the total radiation received by the instrument. When $\alpha = 0°$, the LOSs of measurements will overlap with





each other in the orbital plane. Fig. 11 shows an example of the central measurement track for a vertical sampling of 3 km and horizontal sampling of 165 km projected onto the background atmosphere. Such an observation strategy allows for the retrieval of GWs in a real 2-D space. Assuming the same atmospheric condition and viewing geometry, Song et al. (2017) demonstrated that a GW with a horizontal wavelength longer than 150 km can be retrieved if the horizontal sampling distance

is assumed to be 70 km. The actual horizontal sampling distance is 40 km in our case, and therefore the shortest horizontal wavelength that can be detected is expected to be ∼85 km. Note that the detectable along-track horizontal wavelength in the two cases ($\alpha = 0°$ and $\alpha = 90°$) has a similar limit of ∼85 km, but is derived from different approaches. When $\alpha = 90°$, the horizontal wavelength is indirectly deduced from 1-D vertical profiles. However, a full 2-D wave structure can be obtained from tomographic retrieval approach while $\alpha = 0°$. In Fig. 12, an example of retrieving a GW with a vertical wavelength of 10

km and horizontal wavelength of 150 km is given. In this example, the atmospheric temperature is retrieved in a finer grid: 500 m in the vertical and 7 km in the along-track direction. The 2-D wave structure along the orbital plane can be observed clearly. Additionally, the influence of the across-track variations should also be taken into consideration when the wave is resolved in the along-track direction. As in this case the retrieval assumes a homogeneous atmosphere in the across-track direction, the resolved temperature fields in the along-track direction will be affected by ignoring the across-track variations. For example,

a turning angle of 45° and 90° have different weighting functions in the across-track direction. Thus, the temperature fields in the along-track direction that are reconstructed from them will differ from each other. In Fig. 13, a GW is simulated with a vertical wavelength of 15 km and a horizontal wavelength of 425 km. The wave vector is tilted at an angle of 32° away from the flight direction. Fig. 13 (a) and (b) show the reconstructed wave structures in the along-track direction when turning angles of 45° and 90° are used, respectively. It is obvious that a turning angle of 45° causes a stronger phase shift compared with a

turning angle of 90°. Meanwhile, the wave amplitudes in Fig. 13 (b) are closer to the simulated wave amplitude, which is 10 K. Since in this simulation the wave vector is 32° away from the orbit, a turning angle closer to this wave vector will be affected more by the variations along the LOS. Thus, the wave patterns reconstructed from Fig. 13 (b) fits better with the simulated wave.

## 5.2 Across-track horizontal wavelength

From the retrieval results in Sect. 4.3, we can see that the across-track horizontal wavelength retrieval is affected by the 'sweep mode' turning angle in two aspects. First, the horizontal coverage in the across-track direction is limited by this turning angle. When $\alpha = 45°$, the tangent points from individual limb sounding measurements could extend to ∼1700 km apart from the orbital plane. A larger value of this turning angle will increase the across-track distance that the tangent points are able to reach. Second, the across-track horizontal resolution which determines the smallest detectable wavelength of a wave, is also

affected by this turning angle. As the across-track horizontal resolution increases with distance, smaller waves can be observed better in the region further away from the orbit. This means GWs with smaller horizontal wavelengths in the across-track direction can be observed if the satellite is operated in the 'sweep mode' with a larger turning angle.

Fig. 14 shows an example of retrieving a GW in the across-track direction with a turning angle $\alpha$ of 90°. In this example, the horizontal coverage in the across-track direction extends to 2200 km because a large value of $\alpha$ is selected. Between ∼400-




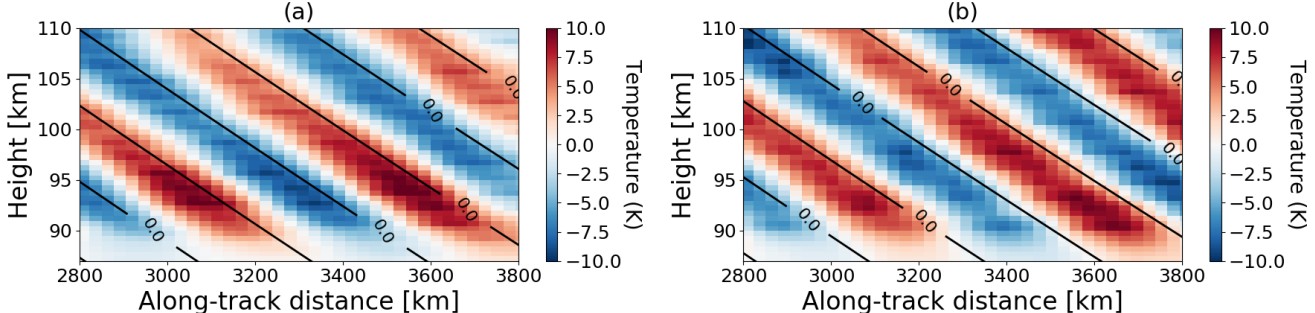

**Figure 13.** Comparison of wave structured retrieved in the along-track direction using a turning angle of $45°$ (a) and $90°$ (b). The wave is simulated with a vertical wavelength of 15 km and a horizontal wavelength of 425 km. The wave vector is tilted at an angle of $32°$ away from the flight direction. The simulated wave amplitude is 10 K. The phase fronts with zero value of the simulated wave are indicated by the black lines.

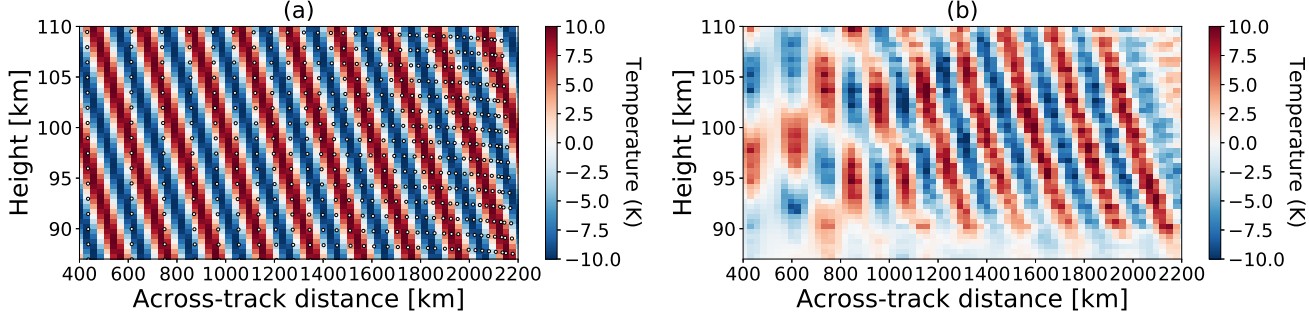

**Figure 14.** Simulated GW (a) with a vertical wavelength of 10 km and horizontal wavelength of 250 km, and retrieved wave structure (b) assuming a turning angle $\alpha$ of $90°$. The tangent points from individual measurements are marked as white circles in (a).

1000 km in the across-track direction, the wave structure can barely be seen as the sampling distance is relatively coarse ($\sim$200 km). In the region close to $\sim$2200 km the tangent points are densely distributed in the across-track direction, and therefore GWs with a small horizontal wavelength can be resolved.

Based on the discussion above, Fig. 15 gives the result associated with its ability in resolving horizontal wavelengths under

5   different viewing modes. In the along-track direction, the smallest horizontal wavelength that can be resolved is constant ($\sim$85 km) because the sampling distance in this direction is independent of the turning angle. In the across-track direction, the resolvable horizontal wavelength is $\sim$850 km at a turning angle of $10°$. GWs with a smaller across-track horizontal wavelength can be resolved if a larger turning angle is adopted. When the turning angle is $45°$, the resolvable horizontal wavelength in both directions reaches the same limit of $\sim$85 km.



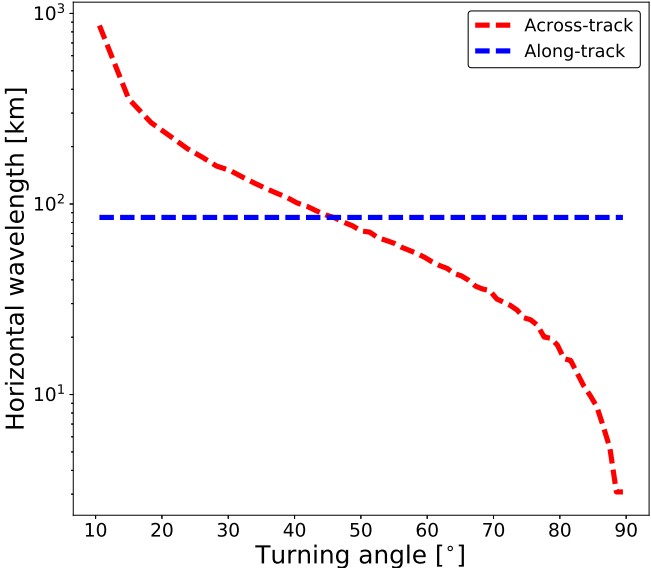

**Figure 15.** Comparison of the ability in resolving horizontal wavelength along the two directions under different turning angles. The blue line represents the minimal detectable horizontal wavelength in the along-track direction, and the red line represents that in the across-track direction. This sensitivity curve is obtained based on the simulation of idealized atmosphere and observation.

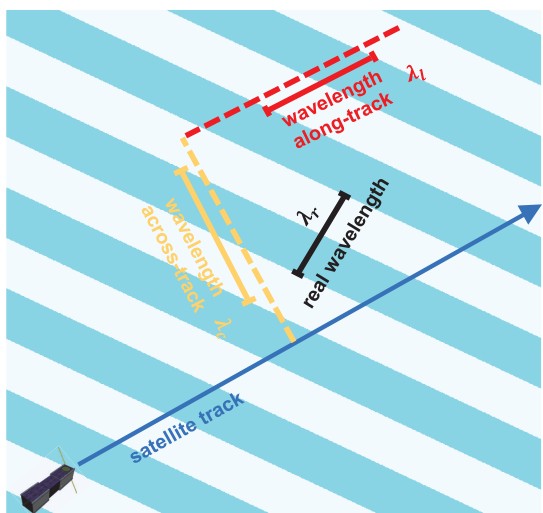

**Figure 16.** Schematic drawing of observed horizontal wavelength in the 'sweep mode' observation. The wave fronts of the GW are indicated by the light-blue shading. The satellite track is represented by the dark-blue arrow. The retrieved horizontal wavelengths in the along- and across-track directions are represented by yellow and red bars, respectively. The real horizontal wavelength is represented by the black bar.





### 5.3 Horizontal wavelength estimation

From conventional limb sounding measurements, the resolved horizontal wavelength is the apparent wavelength projected along the orbital track. This projected horizontal wavelength is generally larger than the real wavelength of a wave, details see in Ern et al. (2004) and Trinh et al. (2015). Utilizing this 'sweep mode' observation strategy, projected wavelengths can be

retrieved along- and across the orbit track in two independent slices. This retrieval scheme could reduce the bias for the estimation of the real wavelength of a wave. The relationship between the retrieved horizontal wavelengths and the real wavelength is shown in Fig. 16. The satellite track is represented by the dark-blue arrow. The wave fronts of the GW are indicated by the light-blue shading. The retrieved horizontal wavelength in the along-track direction ($\lambda_l$) is represented by the red bar, and that in the across-track direction ($\lambda_c$) is represented by the yellow bar. However, the real horizontal wavelength of this simulated

wave is the one that is represented by the black bar. In any case, the real horizontal wavelength of a wave can be derived by combining the horizontal wavelengths retrieved in the two directions perpendicular to each other. The main advantage of this observation mode is that the accuracy of retrieved horizontal wavelengths is independent of the angle between the satellite track and horizontal wave vector. Following a geometric transformation, an unbiased horizontal wavelength estimation can be obtained using this equation:

$$\lambda_h = \frac{\lambda_l \, \lambda_c}{\sqrt{\lambda_l^2 + \lambda_c^2}} \qquad (11)$$

### 6 Conclusion

3-D tomographic reconstruction technique has been used in airborne instruments in recent years. Studies for the satellite mission concept PREMIER have also demonstrated the feasibility of resolving fine GW structures with tomographic retrievals. In this work, a 'sweep mode' observation strategy, which can be used by spaceborne limb sounding instruments for the retrieval

of full 3-D wave vectors in the middle atmosphere, is presented.

An end-to-end simulation of this observation strategy is executed for an instrument measuring the $O_2$ A-band airglow emissions in the MLT region. Employing this approach, two projected 2-D wave structures can be retrieved in the direction along- and across the orbital track, respectively. The turning angular velocity of the instrument needs to be adapted in a way such that the two vertical slices are exactly perpendicular to each other. This special case can be realized by the expected capabilities of the

envisaged satellite platform. The spatial resolution and coverage of the retrieved wave structures in both directions are related with the turning angle of this observation mode, and have been analyzed in the paper. The smallest horizontal wavelength that can be resolved in the along-track direction is constant ($\sim$85 km) for all turning angles, whereas decreases with increasing turning angle in the across-track direction. The wave vectors retrieved in the two directions are then used to deduce the real horizontal wavelength of a wave. This reconstruction technique is independent of the angle between the orbital track and wave

fronts. It provides an unbiased estimation of the real horizontal wavelength of GWs, which is an important parameter for the calculation of GW momentum flux.



The 'sweep mode' as performed here is a large improvement beyond the capabilities of conventional limb sounders that observe with a LOS limited to a fixed azimuth.

**Acknowledgements**

This work was supported in part by the National Natural Science Foundation of China under grant 41590852 and the China

5    Scholarship Council (201404910513).





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
