# Peer review of "3-D tomographic reconstruction of atmospheric gravity waves in the mesosphere and lower thermosphere (MLT)"

_Atmospheric Measurement Techniques, 2017_

## Referee Comment (RC1) · Anonymous Referee #1 · 6 Feb 2018

This paper outlines simulations of retrieving 3D gravity wave structure in the MLT from a satellite platform measuring O2 A-band airglow. It is an interesting topic and approach and is well suited for publication in AMT. Also, it is well written with only a few minor technical issues that need to be addressed (listed below). Some other minor issues (listed below) need to be addressed, mostly for completeness and better clarity. After these issues are properly addressed I would recommend publication.

P2 Lines 16-30 – SABER on TIMED has also been used for a large number of studies on gravity waves. Please include/describe some example references. (e.g. doi:10.1002/2017JD026604, doi:10.1029/2008GL037054, doi: 10.1007/s00382-012-

1329-9, etc.)

P3 Figure 1 –I find the represented geometry in this figure confusing, I think mainly because the instrument LOS is in the middle of the temperature structure (so, we don't know if this is ascending or descending), and it's not clear where the orbital track is. Please include a clearer figure, or perhaps not include this figure as it's not entirely necessary given Fig 2.

P4 Lines 20-21 – Technically, the separate R and P branches follow a Boltzmann distribution (not the entire A-band spectrum), and only when in local thermodynamic equilibrium (and even then not a strict Boltzmann distribution).

P5 Figure 2 – For further clarity, please add black arrows to panel (b), and caption should explain the significance of the purple box. Also, both of these panels seem to be inconsistent with the simulation explanation in Section 4. Please add a panel with the simulation geometry going from 0 to 45°.

P6 Line 8 – airglow is misspelt P7 Eqn 1 – just after the eqn it should be mentioned that O2(1$\Sigma$) in the v=1 state is quickly quenched into the v=0 state, which is why it is necessary to model absorption in the B-band and the subsequent quenching to O2(1$\Sigma$, v=0).

Line 14 – "more than 12 sec" should be "approximately 12 sec".

Line 14 – at higher altitudes (lower pressures), the LTE assumption is going to break down. Have you done any calculations to determine at what altitude LTE can't be assumed? I know the OSIRIS team believes this to be around 110 km, but I can't find any reference to that.

Lines 17-18 – The emission spectrum doesn't simply follow a Boltzmann distribution, and the Mies 1974 reference details how to calculate OH Mienel band transition intensities, which is not the same as the O2 A-band emission spectrum. More discussion/detail is needed here on how you're calculating the A-band emission intensities,

and a better reference would be Babcock and Herzberg 1948 (doi:10.1086/145062)

Line 19 – "aiming to the derivation of" should be "aiming to derive"

Line 21 – please indicate that this is HITRAN 2012

Line 23 – "rotational structure" should be "rotational line emission intensity"

P8 Table 1 – please include references for where the constants were obtained

P9 Line 2 – Need to indicate that you're ignoring other sources of light (i.e. stray light, scattered sunlight) and you're ignoring pressure broadening. Or you could use a more general term in the eqn for the line shape, and then later explain that you can simplify the line shape to be D (which is essentially what you do later anyway).

Eqn 5 – should the "-s" in the second integral be "s"?

Lines 21-22 – please quantify what degree of accuracy you are deeming to be sufficient

Line 23 – "very" is unnecessary

Lines 24-25 – This sentence is very misleading. It seems like you're saying that the only factor determining the airglow emission rate is temperature, and the only factor determining the amount of absorption is O2 density. The emission rate, as detailed in section 3 is dependent on many factors, including both O2 density and temperature; and, as shown in eqns 5, 6, and 7, the self-absorption is dependent on both O2 density and temperature. Please confirm that all relevant processes are being accounted for in your calculations of $\Delta I/I$ (i.e. $\Delta T$ is considered in self-absorption, and $\Delta O2$ is considered in emission rate).

P10 Line 2 – Two or three sentences are needed here giving the basic details of the retrieval algorithm. E.g. Is it global-fitting least squares? MAP? Levenberg-Marquardt? Any a posteriori regularization? Etc.

Lines 7-8 – why are approximate values for the orbit height and tangent altitudes given?

[Figure]

Do they change throughout the simulation? If so, please mention that in the text.

P11 Figure 6 – As previously mentioned, it would be helpful if this figure was consistent with Fig 2. It would also be helpful to have Figs 6 and 7 as two panels in one figure.

Line 7 – "This viewing angle keeps increasing until the predefined turning angle has been reached." was just said in previous sentence

P12 Line 15 – Should "calculating" be "calculation"?

P13 Figure 8 – Unit labels for b-d and f-h should be ∆Temperature (same in Figs 12, 13, and 14). In the first line of the caption, "retrieval results" should be "retrieval a priori, simulated true state, and results"

P14 Figure 9 – Please show results for full altitude range. Or at least explain in the text why only this altitude range is being shown.

Line 4 – What does this profile represent? Is it a single profile at one location, an across-track average, along-track average, total average?

Line 5 – Please explain how gridding error is determined.

Lines 11-12 – Please provide details of how Sa and Se are determined.

P17 Line 19 – "It is obvious" is unnecessary

Line 20 – It would be clearer to discuss "the 90° case" rather than "Fig. 13(b)". Similarly, please specify what it is closer than.

Line 22 – better than what?

P18 Line 9 – What is the minimum wavelength achievable (across-track at 90)?

P20 Line 27 – "whereas it decreases"

---

## Referee Comment (RC2) · Anonymous Referee #2 · 19 Feb 2018

This paper by Rui Song et al outlines a technique for retrieving 3D gravity wave structure in the MLT measuring O2 A-band airglow using an observation strategy that sweeps the line of sight of the limb sounder horizontally across the orbotal track during flight from a nano-satellite (cubesat) platform. I find the paper well written, the literature review comprehensive, the sections are being logically organised and the simulations and retrievals presented compelling to illustrate the technique. I recommend I paper be accepted and only offer the following suggestions for consideration by the authors. 1) The literature review can be strengthened by adding some references of gravity wave detection from SABER instrument onboard TIMED satellite, 2) GW detection from the SOFIE instrument onboard the AIM satellite and GW detection from the CIPS instrument which images gravity waves in PMC's with horizontal wavelengths not detected by any other instrument in the MLT. 2) Figure 1 is confusing and can use a better figure to illustrate the sweep mechanism along the line of sight strategy. 3) This consideration is probably beside the scope of the paper, but the paper can be improved by a discussion on stray light considerations for the instrument and stray light effects on retrievals. A discussion on the radiances required for accurate estimation of GW parameters from the O2 airglow emission may be included. How do errors in slew rate, positioning accuracy and jitter affect the retrievals?

---

## Author Comment (AC1) · 19 Apr 2018

This paper outlines simulations of retrieving 3D gravity wave structure in the MLT from a satellite platform measuring O2 A-band airglow. It is an interesting topic and approach and is well suited for publication in AMT. Also, it is well written with only a few minor technical issues that need to be addressed (listed below). Some other minor issues (listed below) need to be addressed, mostly for completeness and better clarity. After these issues are properly addressed I would recommend publication.

We thank the referee for carefully reviewing the manuscript and for the positive

comments.

P2 Lines 16-30 – SABER on TIMED has also been used for a large number of studies on gravity waves. Please include/describe some example references. (e.g. doi:10.1002/2017JD026604, doi:10.1029/2008GL037054, doi: 10.1007/s00382-012-1329-9, etc.)

Thanks for the suggestion. The work on SABER and relevant references have been added:
"...The Sounding of the Atmosphere using Broadband Emission Radiometry (SABER) measures temperature between 20 and 110 km. The observations are sensitive to GWs with horizontal and vertical wavelengths longer than ∼100-200 km and ∼4 km (Preusse et al., 2002; Schroeder et al., 2009; John and Kumar, 2012; Liu et al., 2017)..."

P3 Figure 1 –I find the represented geometry in this figure confusing, I think mainly because the instrument LOS is in the middle of the temperature structure (so, we don't know if this is ascending or descending), and it's not clear where the orbital track is. Please include a clearer figure, or perhaps not include this figure as it's not entirely necessary given Fig 2.

Thanks for the suggestion. We have removed Figure 1 from the manuscript. The observation strategy is illustrated entirely based on Figure 2 now.

P4 Lines 20-21 – Technically, the separate R and P branches follow a Boltzmann

distribution (not the entire A-band spectrum), and only when in local thermodynamic equilibrium (and even then not a strict Boltzmann distribution).

The text has been revised to clarify this. "The P and R branch emission lines follows a Boltzmann distribution described by the kinetic temperature under the assumption of local thermodynamic equilibrium (LTE)."

P5 Figure 2 - For further clarity, please add black arrows to panel (b), and caption should explain the significance of the purple box. Also, both of these panels seem to be inconsistent with the simulation explanation in Section 4. Please add a panel with the simulation geometry going from 0 to 45°.

Thanks for the comment. A black arrow was added in panel (b), representing the viewing direction of the instrument. The purple box was further explained:
"In panel (b), the purple box represents the region where the 'sweep mode' takes place."
In this figure, we used the simulation geometry going from 10 to 90° as the example to illustrate the 'sweep mode'. We also tried to plot the case of 0 to 45°. However, it is very misleading when the viewing angle is 0° and the tangent points overlap with the satellite track. Therefore, we used the 10° as the initial viewing angle. The 90° viewing angle could produce a larger region for tomographic retrieval, and therefor used in this plot for the geometric illustration.

P6 Line 8 – airglow is misspelt.
Corrected

P7 Eqn 1 – just after the eqn it should be mentioned that O2(1Σ) in the v=1 state is quickly quenched into the v=0 state, which is why it is necessary to model absorption in the B-band and the subsequent quenching to O2(1Σ, v=0).

A sentence is added. "The B-band is considered due to the fact that $O_2(b^1\Sigma)$ in the $v = 1$ state is rapidly collisionally deactivated into the $v = 0$ state, subsequently leading to A-band emission."

Line 14 – "more than 12 sec" should be "approximately 12 sec"
Corrected

Line 14 – at higher altitudes (lower pressures), the LTE assumption is going to break down. Have you done any calculations to determine at what altitude LTE can't be assumed? I know the OSIRIS team believes this to be around 110 km, but I can't find any reference to that.
We don't have a calculation in this aspect. We use the same LTE assumption as the OSIRIS team's work.

Lines 17-18 – The emission spectrum doesn't simply follow a Boltzmann distribution, and the Mies 1974 reference details how to calculate OH Mienel band transition intensities, which is not the same as the O2 A-band emission spectrum. More discussion/detail is needed here on how you're calculating the A-band emission intensities, and a better reference would be Babcock and Herzberg 1948 (doi:10.1086/145062).
Detailed steps for calculating the A-band emission spectrum were given in our previous

publication: Song et al., 2017, doi: 10.5194/amt-2017-118. Here, a new sentence with right references was added:
Babcock and Herzberg (1948) discussed the fine structure of $O_2$ A-band emission in details, and equations for calculating the emission spectrum were summarized in Song et al. (2017) from Eq. 4 - Eq. 5.

Line 19 – "aiming to the derivation of" should be "aiming to derive".
Corrected.

Line 21 – please indicate that this is HITRAN 2012.
Corrected.

Line 23 – "rotational structure" should be "rotational line emission intensity".
Corrected.

P8 Table 1 – please include references for where the constants were obtained.
A reference of HITRAN 2012 was added.

P9 Line 2 – Need to indicate that you're ignoring other sources of light (i.e. stray light, scattered sunlight) and you're ignoring pressure broadening. Or you could use a more general term in the eqn for the line shape, and then later explain that you can simplify the line shape to be D (which is essentially what you do later anyway).

A sentence was added at the end of this paragraph:
"The pressure broadening and other sources of light (i.e. stray light, scattered sunlight) are ignored in this case."

Eqn 5 – should the "-s" in the second integral be "s"?
Corrected.

Lines 21-22 – please quantify what degree of accuracy you are deeming to be sufficient.
A quantitative description is added.
"The difference between adaptive grid-spacing and regular grid-spacing in spectrum-integrated intensity is less than 1."

Line 23 – "very" is unnecessary.
"Very" is removed.

Lines 24-25 – This sentence is very misleading. It seems like you're saying that the only factor determining the airglow emission rate is temperature, and the only factor determining the amount of absorption is O2 density. The emission rate, as detailed in section 3 is dependent on many factors, including both O2 density and temperature; and, as shown in eqns 5, 6, and 7, the self-absorption is dependent on both O2 density and temperature. Please confirm that all relevant processes are being accounted for in your calculations of $\Delta I/I$ (i.e. $\Delta T$ is considered in self-absorption, and $\Delta O2$ is considered in emission rate).

In the calculations of $\Delta I(v)/I(v)$, all the relevant processes were being accounted for. Here we reorganized the sentence:
"The temperature and $O_2$ can determine the airglow emission rate, as well as the amount of self-absorption."

P10 Line 2 - Two or three sentences are needed here giving the basic details of the retrieval algorithm. E.g. Is it global-fitting least squares? MAP? Levenberg-Marquardt? Any a posteriori regularization? Etc.
Two sentences are added to describe the adopted retrieval and regularization scheme.
"The Levenberg-Marquardt iteration method is used to minimize the cost function of the non-linear problem. The Tikhonov regularization is used to ensure that a unique and physically meaningful solution can be obtained."

Lines 7-8 - why are approximate values for the orbit height and tangent altitudes given? Do they change throughout the simulation? If so, please mention that in the text.
The orbit height and tangent altitudes are fixed values. The "$\sim$" has been deleted.

P11 Figure 6 – As previously mentioned, it would be helpful if this figure was consistent with Fig 2. It would also be helpful to have Figs 6 and 7 as two panels in one figure.
The two figures have been combined into one figure.

Line 7 – "This viewing angle keeps increasing until the predefined turning angle has been reached." was just said in previous sentence.
This sentence has been removed.

P12 Line 15 – Should "calculating" be "calculation"?
Corrected.

P13 Figure 8 – Unit labels for b-d and f-h should be △Temperature (same in Figs 12, 13, and 14). In the first line of the caption, "retrieval results" should be "retrieval a priori, simulated true state, and results"
Unit labels for Fig. 8 (b-d, f-h), Fig. 12, 13 and 14 have been corrected as △Temperature. In the caption, the "retrieval results' was replaced as "retrieval a priori, simulated true state, and results".

P14 Figure 9 – Please show results for full altitude range. Or at least explain in the text why only this altitude range is being shown.
A sentence has been added to explain why only this altitude range is shown.
"Temperature retrievals below 87 km and above 110 km are excluded from the results because of the decreased sensitivity to small temperature perturbations."

Line 4 – What does this profile represent? Is it a single profile at one location, an across-track average, along-track average, total average?
Figure. 9 shows a single vertical profile of retrieved temperature at one location and the total deviation from the expected profile...

Line 5 – Please explain how gridding error is determined.

The gridding error represents the error induced by the discretization of the retrieval grid. This explanation was added in the bracket.

Lines 11-12 – Please provide details of how Sa and Se are determined.
Details of how Sa and Se are determined has been added.
$\mathbf{S}_a^{-1}$ is the regularization matrix used to constrain the solution. Here, we used a combination of zeroth- and first-order Tikhonov regularization (Tikhonov and Arsenin, 1977). $\mathbf{S}_\epsilon^{-1}$ is the covariance matrix of measurement errors, and assumed to be diagonal.

P17 Line 19 – "It is obvious" is unnecessary.
"It is obvious that" is deleted.

Line 20 – It would be clearer to discuss "the 90∘ case" rather than "Fig. 13(b)". Similarly, please specify what it is closer than.
This sentence has been rewritten.
"Meanwhile, the wave amplitudes in the 90° case are closer to the simulated wave amplitude than in the 45° case."

Line 22 – better than what?
This sentence has been rewritten.
"Thus, the reconstructed wave in Fig. 13 (b) fits better with the simulated wave than in Fig 13 (a)."

P18 Line 9 – What is the minimum wavelength achievable (across-track at 90)?
A sentence that describes the achievable minimum wavelength was added.
"For an idealized atmosphere and idealized observation, horizontal wavelengths in the

across-track direction down to a few km can be resolved."

P20 Line 27 – "whereas it decreases"
Corrected.

---

## Author Comment (AC2) · 19 Apr 2018

This paper by Rui Song et al outlines a technique for retrieving 3D gravity wave structure in the MLT measuring O2 A-band airglow using an observation strategy that sweeps the line of sight of the limb sounder horizontally across the orbotal track during flight from a nano-satellite (cubesat) platform. I find the paper well written, the literature review comprehensive, the sections are being logically organised and the simulations and retrievals presented compelling to illustrate the technique. I recommend I paper be accepted and only offer the following suggestions for consideration by the authors.

[Figure]

We thank the reviewer for providing a thorough review and offering valuable suggestions.

The literature review can be strengthened by adding some references of gravity wave detection from SABER instrument onboard TIMED satellite, 2) GW detection from the SOFIE instrument onboard the AIM satellite and GW detection from the CIPS instrument which images gravity waves in PMC's with horizontal wavelengths not detected by any other instrument in the MLT.

The work on SABER and relevant references have been added:

"...The Sounding of the Atmosphere using Broadband Emission Radiometry (SABER) measures temperature between 20 and 110 km. The observations are sensitive to GWs with horizontal and vertical wavelengths longer than ∼100-200 km and ∼4 km (Preusse et al., 2002; Schroeder et al., 2009; John and Kumar, 2012; Liu et al., 2017)..."

Reference of the SOFIE instrument was added:

"The Solar Occultation for Ice Experiment (SOFIE) onboard the Aeronomy of Ice in the Mesosphere (AIM) satellite measures temperature profiles in the height range of 10-102 km in the Arctic and Antarctic. The SOFIE measurements can be used to study seasonal and annual variations of GWs in the entire polar stratosphere and mesosphere in both hemispheres (Liu et al., 2014). Compared to limb sounding or occultation measurements, ..."

Reference of the CIPS instrument was added:

"The Cloud Imaging and Particle Size(CIPS) instrument onboard the AIM satellite measures polar mesospheric cloud (PMC) morphology and particle properties. GWs derived from nadir viewing of CIPS have horizontal wavelengths mainly in the range of 250-300 km (Chandran et al., 2010)."

Figure 1 is confusing and can use a better figure to illustrate the sweep mechanism along the line of sight strategy.

Thanks for the suggestion. The other referee also thinks this figure is confusing, and suggests to not include this figure as it's not entirely necessary given Fig. 2. Therefore, we removed Fig. 1 from the updated manuscript.

This consideration is probably beside the scope of the paper, but the paper can be improved by a discussion on stray light considerations for the instrument and stray light effects on retrievals. A discussion on the radiances required for accurate estimation of GW parameters from the O2 airglow emission may be included. How do errors in slew rate, positioning accuracy and jitter affect the retrievals?

In this work, we assumed the stray light can be suppressed very well by the instrument. Here, we added a sentence in Sect. 3.3:

"... and other sources of light (i.e. stray light, scattered sunlight) are ignored in this case."

In our method, the GW parameter and temperature retrieval relies on the relative intensity of the emission lines, not the absolute airglow emissions. In Sect.4.3, we obtained a temperature retrieval precision of 2 K at altitudes between 87 and 110 km.

A sentence was added in Sect.4.1 to explain the errors in the slew rate:

"Considering a $L$ of $\sim$ 2600 km, an $1°$ error in the turning angle can result in a $\sim$45 km offset of the tangent points in the horizontal plane."
* * *